# Multi-task adaptive deep sparse canonical correlation analysis for multi-omics cancer survival prediction

Yan Wang[1]*, Zimo Zou[2], Yuanyuan Wu[1], Jing Chen[3]

1 College of Life Sciences, Xinyang Normal University, Xinyang, Henan, China, 2 Department of Electrical, Computer and Software Engineering, University of Auckland, Auckland, New Zealand, 3 College of International Education, Xinyang Normal University, Xinyang, Henan, China

* wangyan0508@xynu.edu.cn

## Abstract

Integrating high-dimensional multi-omics data is essential for uncovering the coordinated molecular mechanisms underlying cancer progression and improving survival prediction. DNA methylation and mRNA expression represent two tightly coupled regulatory layers; however, many existing approaches either model them independently or rely on linear assumptions that fail to capture the nonlinear cross-omics structure. Here, we propose MT-ADSCCA, a multitask adaptive deep sparse canonical correlation analysis framework that jointly learns correlated latent representations, selects interpretable multi-omics biomarkers, and supports downstream survival modeling. MT-ADSCCA embeds sparse CCA into a nonlinear encoder architecture and uses uncertainty-guided adaptive weighting to stabilize multi-objective training. The selected features were subsequently modeled using a BiLSTM–Cox survival network with genes ordered by chromosomal coordinates to capture local genomic dependencies. We evaluated MT-ADSCCA using event-stratified nested 10-fold cross-validation across three TCGA cohorts: breast invasive carcinoma (BRCA), glioma (GBMLGG), and pan-kidney carcinoma (KIPAN), including 485, 563, and 652 matched multi-omics samples, respectively. MT-ADSCCA achieved the highest concordance indices across all cohorts, outperforming six feature-selection baselines (DA, WGCNA, lmQCM, CCA, OSCCA, DeepCorrSurv) and four survival-model baselines (LASSO-Cox, RSF, MTLSA, DeepSurv). Kaplan–Meier analyses further confirmed a clear separation between the predicted high- and low-risk groups. The selected canonical features were enriched in biologically coherent functional categories, supporting the interpretability of the learned patterns. Together, these results demonstrate that MT-ADSCCA provides a robust and interpretable framework for multi-omics integration and cancer prognosis prediction.

**Data availability statement:** All relevant data are within the manuscript and its Supporting information files.

**Funding:** Nanhu Scholars Program for Young Scholars of XYNU.

**Competing interests:** The authors have declared that no competing interests exist.

## Introduction

Cancer remains a leading cause of mortality worldwide and presents substantial biological and clinical heterogeneity that complicates diagnosis, treatment selection, and prognostic assessment [1]. With the rapid expansion of large-scale genomic consortia, such as The Cancer Genome Atlas (TCGA), multiple layers of molecular information, including gene expression, DNA methylation, copy-number variation, proteomics, and somatic mutations, have become widely available for clinical and translational research [2]. These multi-omics datasets offer unprecedented opportunities to characterize the complex regulatory networks underlying tumor development and progression. However, capitalizing on these data for robust survival prediction remains a major computational challenge, largely due to the high dimensionality, strong noise, and nonlinear relationships across omics layers.

Among the numerous molecular modalities, DNA methylation and mRNA gene expression represent two of the most biologically interconnected. DNA methylation, particularly in promoter regions, directly regulates transcriptional activity and influences tumor subtype determination, therapy response, and disease aggressiveness. Conversely, mRNA expression reflects the downstream phenotypic outputs of epigenetic and transcriptional regulation. Therefore, integrating these two modalities has strong potential to uncover cross-omics biomarkers with mechanistic and prognostic significance [3]. Nevertheless, naïve concatenation or independent modeling of methylation and expression often fails to capture the coordinated biological signals shared across the layers.

Canonical correlation analysis (CCA) and its sparse variants (SCCA) provide a statistical framework for learning correlated projections from paired datasets and have been widely used in the field of genomics. Sparse CCA, in particular, encourages simultaneous dimensionality reduction and gene selection, making it attractive for high-dimensional multi-omics studies [4–6]. However, traditional sparse CCA remains limited by its reliance on linear relationships, sensitivity to noise, and difficulty in tuning the sparsity parameters [7,8]. These limitations restrict its ability to detect complex nonlinear dependencies between methylation and gene expression, which are well-characterized in cancer biology.

Recent advances in deep learning have prompted the development of nonlinear correlation learning frameworks, such as DeepCCA [9], and related multiview neural architectures. These models can capture nonlinear cross-omics representations but typically lack interpretability and require large sample sizes, which are often unavailable in clinical genomics [10]. Additionally, most deep correlation models do not directly incorporate survival outcomes during representation learning, missing the opportunity to identify biomarkers that are both cross-modally correlated and clinically relevant [11].

In light of these challenges, developing integrative models that can jointly capture shared cross-omics structures, enforce interpretability, and incorporate survival outcomes remains an important open problem. Although sparse canonical correlation analysis (SCCA) provides a principled framework for multi-omics integration, traditional formulations suffer from several well-recognized limitations when applied to

cancer genomics [12]. First, classical SCCA relies strictly on linear projection functions, which are inadequate for representing nonlinear interactions, such as immune–epigenetic interactions, chromatin remodeling, ECM-related signaling, and metabolic or transport-related regulation observed in cancer biology. These nonlinear processes are ubiquitous in cancer biology and cannot be captured using simple linear combinations of features. Second, SCCA optimization is highly sensitive to hyperparameters that control sparsity. Determining the appropriate strength for L1 penalties often requires extensive grid searches, which are computationally expensive and prone to overfitting when the sample sizes are small. Third, SCCA typically treats canonical correlation learning as independent of clinical outcomes; thus, the model may prioritize features that are correlated but clinically irrelevant [13].

To overcome the linearity and interpretability constraints of SCCA, several nonlinear multiview learning approaches have been proposed. DeepCCA, one of the earliest methods, replaces linear projections with neural networks to capture complex relationships across datasets [14]. Althoug powerful, DeepCCA often requires large training samples, and its fully connected architecture lacks biological interpretability because it treats all features as equally structured and independent. Other nonlinear extensions, such as kernel CCA or autoencoder-based multi-omics fusion models, also struggle with transparency. Their dense representations obscure direct mapping of learned latent variables to specific genes, making it challenging to derive mechanistic insights or clinically actionable biomarkers. This issue has been highlighted by multiple recent studies emphasizing interpretability as an essential requirement for precision oncology.

In addition to the challenge of representation learning, effective feature selection remains a central problem in multi-omics analyses. High-dimensional omics datasets typically contain tens of thousands of features, of which only a small subset is mechanistically or clinically meaningful. Existing filtering-based methods, including weighted gene co-expression network analysis (WGCNA), lmQCM [15,16] clustering, and differential analysis (DA) [17], often operate independently on each omics layer and do not guarantee that the selected features maximize the cross-omics correlation [18]. More advanced approaches, such as OSCCA, introduce structured sparsity but still rely on linear correlations and manually fixed sparsity thresholds. Deep-learning-based models, such as DeepCorrSurv [19], attempt to couple multi-omics correlation learning [19] with survival prediction; however, these methods still face challenges in balancing multiple loss terms, mitigating overfitting, and ensuring robust interpretability.

However, existing approaches rarely unify nonlinear correlation learning, adaptive sparsity, and outcome-aware modeling within a single end-to-end framework. Multi-task learning offers a compelling solution: by simultaneously optimizing correlation objectives, sparsity constraints, reconstruction consistency, and survival relevance, the model can learn richer and more clinically meaningful multi-omics representations [20]. However, standard multi-task learning approaches typically require manually defined loss weights, and poorly chosen weights can destabilize training or skew the model toward one objective at the expense of others [21].

These limitations collectively underscore the need for a methodological advance that combines (1) nonlinear multi-omics representation learning, (2) interpretable sparse biomarker discovery, (3) task-adaptive balancing of multiple learning objectives, and (4) integration with survival prediction models capable of modeling biologically structured gene relationships. Such a framework would not only address the critical weaknesses of existing approaches but also create new opportunities for deriving clinically meaningful insights from complex multi-omics data.

To address the methodological and biological gaps outlined above, we propose the multi-task adaptive deep sparse canonical correlation analysis (MT-ADSCCA) framework for multi-omics integration in cancer prognosis. MT-ADSCCA reformulates sparse CCA as a multi-objective deep learning model, enabling the joint optimization of cross-omics correlation learning, sparse feature selection, and representation regularization within a unified architecture. In addition, MT-ADSCCA adopts an uncertainty-guided adaptive loss weighting to improve training stability and reduce sensitivity to manually tuned loss coefficients.

To model survival outcomes, the selected multi-omics features were subsequently used to train a bidirectional LSTM–Cox survival network, in which genes were ordered by chromosomal position to capture local genomic dependencies. The

validity of this ordering assumption was evaluated using a shuffle test. Finally, we assessed MT-ADSCCA using a rigorous nested cross-validation protocol to ensure unbiased performance estimation and reproducibility across cohorts.

In summary, the contributions of this study are fourfold: (1) a multi-task deep sparse CCA framework that learns correlated nonlinear representations while selecting interpretable biomarker subsets; (2) an uncertainty-based adaptive weighting mechanism for stable multi-objective optimization; (3) a genomically ordered BiLSTM–Cox survival modeling strategy for capturing higher-order gene dependencies; and (4) a comprehensive evaluation on three TCGA cohorts demonstrating consistent performance gains over representative multi-omics feature-selection and survival-prediction baselines.

## Methods

### Study design and overview

This study evaluated MT-ADSCCA, a multi-task adaptive deep sparse canonical correlation framework for integrating DNA methylation and mRNA expression to improve cancer survival prediction and enable biologically interpretable multi-omics biomarker discovery. We benchmarked MT-ADSCCA across three biologically diverse TCGA cohorts: breast invasive carcinoma (BRCA), glioma (GBMLGG), and pan-kidney carcinoma (KIPAN).

### Data sources and cohort construction

We used matched multi-omics and clinical survival datasets from The Cancer Genome Atlas (TCGA) [22]. All datasets were downloaded from the Broad Institute GDAC FireBrowse portal [23], which provides uniformly processed TCGA level-3 multi-omics matrices suitable for reproducible computational analyses.

For each cohort, we collected: (1) mRNA gene expression profiles (RNA-SeqV2 RSEM), (2) DNA methylation profiles (Illumina HumanMethylation450, gene-level aggregated beta values), and (3) clinical survival annotations, including overall survival time and event indicators. Omics profiles and clinical annotations were merged using TCGA barcode prefixes.

The cohort inclusion criteria were applied consistently across BRCA, GBMLGG, and KIPAN. We retained patients with: (i) valid clinical follow-up information, (ii) matched mRNA expression and DNA methylation profiles, and (iii) non-missing values after preprocessing. Overall survival (OS) time was defined as days-to-death for deceased patients and days-to-last-follow-up for censored patients and was converted to months for reporting. Patients with OS time ≤ 0 were excluded from the analysis. The final cohort characteristics are presented in Table 1.

**Table 1. Baseline clinical and molecular characteristics of the three TCGA cohorts.**

| Characteristics | BRCA (n = 485) | GBMLGG (n = 563) | KIPAN (n = 652) |
|---|---|---|---|
| **Age (years), median (IQR)** | N/A | N/A | N/A |
| **Sex (Female/Male)** | N/A | N/A | N/A |
| **Events (deceased)** | 69 (14.2%) | 158 (28.1%) | 153 (23.5%) |
| **Censored (alive)** | 416 (85.8%) | 405 (71.9%) | 499 (76.5%) |
| **Follow-up time (months)** | 0.03–282.69 | 0.03–116.03 | 0.02–197.24 |
| **mRNA gene count** | 20,533 | 17,184 | 25,055 |
| **DNA methylation gene count** | 20,106 | 20,116 | 20,533 |
| **Matched multi-omics samples** | 485 | 563 | 652 |
| **Missing variables** | Age, Sex | Age, Sex | Age, Sex |

Age and sex information were not available in the FireBrowse-processed TCGA multi-omics matrices used in this study. These variables are therefore reported as "N/A". Follow-up time is shown as the observed range because quartile statistics could not be derived from the pre-harmonized dataset. Event and censoring counts were extracted from TCGA clinical annotations.

Additional methodological details and supplementary analyses are provided in Supplementary S1–S5 Tables, including nested cross-validation performance summaries, shuffle-test validation of genomic ordering for the BiLSTM–Cox survival head, DeepSurv baseline configuration, additional multi-omics integration baselines with paired statistical testing and effect-size reporting, and train-versus-validation canonical-correlation stability results.

## Preprocessing and feature harmonization

We harmonized the mRNA and methylation matrices by mapping gene identifiers to standardized HUGO gene symbols and retaining only the genes present in both modalities. For mRNA expression, RSEM values were transformed using $\log_2(RSEM + 1)$, and the z-score was normalized across samples. For DNA methylation, gene-level beta values were z-score normalized across samples.

To reduce noise and computational burden, we applied variance-based filtering independently within each modality by removing genes whose variance fell below the 10th percentile. All preprocessing steps were performed strictly within training folds during nested cross-validation and then applied to the held-out test folds without refitting.

## MT-ADSCCA model formulation

Let $X \in R^{n \times p}$ denote the mRNA expression matrix and $Y \in R^{n \times q}$ denote the DNA methylation matrix, where n is the number of patients and p, q are the numbers of genes in each modality. Canonical correlation analysis (CCA) seeks projection vectors u and v that maximize the correlation between the projected variables $\boldsymbol{Xu}$ and $\boldsymbol{Yv}$.

Classical sparse CCA (SCCA) introduces ℓ1 penalties to encourage interpretability:

$$\max_{u,v} \frac{\boldsymbol{u}^T \boldsymbol{X}^T \boldsymbol{Y} \boldsymbol{v}}{\sqrt{\boldsymbol{u}^T \boldsymbol{X}^T \boldsymbol{X} \boldsymbol{u}} \sqrt{\boldsymbol{v}^T \boldsymbol{Y}^T \boldsymbol{Y} \boldsymbol{v}}} - \lambda_u \| \boldsymbol{u} \|_1 - \lambda_v \| \boldsymbol{v} \|_1$$

However, linear projections are insufficient for capturing nonlinear regulatory interactions, and fixed sparsity penalties require careful manual tuning. MT-ADSCCA addresses these limitations by learning nonlinear encoders while preserving sparse, gene-level canonical weights.

Nonlinear encoders map each modality into a latent representation:

$$\boldsymbol{h_x} = \boldsymbol{f_\theta}(\boldsymbol{X}), \qquad \boldsymbol{h_y} = \boldsymbol{g_\varnothing}(\boldsymbol{Y})$$

where $\boldsymbol{f_\theta}$ and $\boldsymbol{g_\varnothing}$ are multilayer feed-forward networks (2–3 hidden layers, batch normalization, ReLU activations). These encoders map each omics modality into a latent representation that captures nonlinear patterns.

To maintain interpretability, MT-ADSCCA applies **sparse canonical weights** directly on the encoded features:

$$z_x = h_x \circ u, \qquad z_y = h_y \circ v$$

where "∘" denotes elementwise multiplication. Thus, sparsity is enforced at the final gene-interface layer, preserving transparent mapping between canonical weights and gene-level contributions.

Multi-task objective. MT-ADSCCA optimizes four coupled objectives:

$$L = \lambda_1 \mathcal{L}_{corr} + \lambda_2 \mathcal{L}_{sparse} + \lambda_3 \mathcal{L}_{recon} + \lambda_4 \mathcal{L}_{norm}$$

1) Canonical correlation loss (maximize cross-omics alignment):

$$\mathcal{L}_{corr} = -corr\left(z_x, z_y\right) = -\frac{u^T X^T Y v}{\sqrt{u^T X^T X u}\sqrt{v^T Y^T Y v}}$$

2) Sparsity loss (compact and interpretable biomarker discovery):

$$\mathcal{L}_{sparse} = \|u\|_1 + \|v\|_1$$

3) Reconstruction loss (stabilize representation learning):

$$\mathcal{L}_{recon} = \left\|X - \hat{X}\right\|_2^2 + \left\|Y - \hat{Y}\right\|_2^2$$

4) Normalization loss (prevent degenerate solutions and stabilize gradients):

$$\mathcal{L}_{norm} = \left(Var\left(z_x\right) - 1\right)^2 + \left(Var\left(z_y\right) - 1\right)^2$$

In traditional multi-task learning, fixed weights $\lambda_i$ must be manually tuned, which is difficult for multi-omics applications. MT-ADSCCA instead learns the task weights:

$$\lambda_i = \frac{1}{2\sigma_i^2}$$

where each $\sigma_i$ is a trainable uncertainty parameter (enforced positive via softplus). Task weights are learned using uncertainty-guided multi-task learning (trainable $\sigma_i$), enabling automatic rebalancing of correlation learning, sparsity, reconstruction, and normalization during training.

## Stability-driven feature selection strategy

To identify robust biomarkers, MT-ADSCCA was integrated with a stability-driven feature selection procedure within cross-validation. In each training fold, MT-ADSCCA learned sparse canonical weight vectors **u** and **v**. A gene was considered selected in that fold if it had a nonzero contribution in either modality: $|u_j| > 0$ $or$ $|v_j| > 0$.A

After running MT-ADSCCA across all folds, we computed the selection frequency for each gene and retained only genes selected in at least 9 out of 10 folds. This stability criterion reduces the number of false positives and improves the reproducibility in high-dimensional omics settings. The final stable feature sets consisted of 260 genes for BRCA, 1,003 genes for GBMLGG, and 576 genes for KIPAN, respectively.

## Genomic ordering and BiLSTM survival modeling

**Gene ordering strategy.** After selecting stable multi-omics biomarkers, genes were ordered by chromosome and genomic coordinate. This ordering reflects biological assumptions such as local chromatin organization, co-methylation domains, and coordinated regulation of neighboring genes. A shuffle test validating the importance of genomic ordering is presented in Supplementary S2 Table.

**BiLSTM–Cox survival head.** Each patient was represented by an ordered sequence of multi-omics gene features. A bidirectional long short-term memory (BiLSTM) network was used to process the sequence to learn the local genomic dependencies. The resulting hidden representations were pooled to form a patient-level embedding, which was mapped to a risk score using the Cox layer. Model training used the Cox partial log-likelihood, which naturally accommodates censoring.

Dropout (0.1) and L2 regularization were applied to reduce overfitting. Survival-model baselines were evaluated using the same nested cross-validation protocol for a fair comparison. For reproducibility, the DeepSurv architecture and training hyperparameters used in our experiments are listed in Supplementary S3 Table.

### Training procedure and implementation details

All MT-ADSCCA components were optimized using the Adam optimizer. The initial learning rate was set to 0.001 and decayed by a factor of 0.5 for every 100 epochs. Training was performed for 1000 epochs per fold, consistent with the observed convergence behavior.

To improve training stability, we applied batch normalization in the encoder layers, dropout (0.1) in the survival head, L2 weight regularization ($1e-4$), and gradient clipping with global norm 5. Batch size was set to 32.

### Experimental design and evaluation protocol

All experiments were conducted using a nested 10-fold cross-validation (CV) protocol to prevent information leakage and ensure unbiased generalization estimates.

In the outer loop, one-fold was held out as the test set for final evaluation. In the inner loop, all model training, hyperparameter selection, and feature selection were performed using only the training data. Importantly, all preprocessing steps (including normalization, variance filtering, and MT-ADSCCA feature selection) were fitted on the training folds and applied to the test folds without refitting.

Because survival datasets are intrinsically imbalanced due to censoring, we used event-stratified splitting based on the event indicator (deceased vs censored), ensuring that each fold preserved the cohort-level event proportion.

### Evaluation metrics

The primary evaluation metric was the concordance index (C-index), which measures the proportion of concordant risk–time pairs and is robust to censoring. Secondary evaluation metrics included Kaplan–Meier (KM) stratification, log-rank test p-values, and hazard ratios (HRs) with 95% confidence intervals. For KM analysis, patients were dichotomized into high- and low-risk groups using the median predicted risk score within each test fold.

To assess representation stability, we also tracked train-versus-validation canonical correlations across folds (Supplementary S5 Table).

### Statistical significance testing

To compare MT-ADSCCA with baseline models, we conducted paired Wilcoxon signed-rank test on per-fold C-index values. This nonparametric paired test is appropriate for cross-validation-based survival-model comparisons. Bonferroni correction was applied to adjust for multiple comparisons within each cohort.

We also reported effect sizes (Cohen's d) [24] to quantify the magnitude of performance differences across methods.

## Results

### Cohort overview and evaluation strategy

We evaluated the performance of MT-ADSCCA on three biologically diverse TCGA cohorts (BRCA, GBMLGG, KIPAN). Although these datasets differ in disease aggressiveness and follow-up length, their heterogeneity provides a rigorous

testbed for multi-omics survival modeling. BRCA includes long-term survivors with substantial censoring; GBMLGG exhibits rapid progression with shorter follow-up; and KIPAN represents an intermediate profile. These contrasts enable a comprehensive assessment of model robustness across distinct survival-time distributions.

## Overall performance and statistical significance

To ensure that the observed performance gains were not driven by a small number of favorable splits, we performed paired statistical testing on the per-fold C-index values under the nested cross-validation protocol. For each cohort, MT-ADSCCA was compared against each baseline using a paired Wilcoxon signed-rank test, and p-values were adjusted using the Bonferroni correction within each cohort. In addition, we reported effect sizes (Cohen's d) to quantify the magnitude of improvement. The overall predictive performance under nested cross-validation is summarized in Supplementary S1 Table, and detailed statistical testing results are provided in Supplementary S4 Table.

## Comparison with multi-omics feature-selection baselines

We compared MT-ADSCCA against six feature-selection and multi-omics integration baselines, including DA, WGCNA, lmQCM, classical CCA, OSCCA, and DeepCorrSurv. For each baseline, mRNA expression and DNA methylation features were selected independently and concatenated, and the same BiLSTM–Cox survival head was trained under identical nested cross-validation folds to ensure a fair comparison. All feature selections were performed strictly within the training folds and applied to the held-out test folds to avoid information leakage.

Across all evaluated datasets, MT-ADSCCA demonstrated statistically significant superiority over the following baseline methods after Bonferroni correction within each cohort ($\alpha = 0.05/6 \approx 0.0083$): DA ($p\_adj < 1 \times 10^{-4}$), WGCNA ($p\_adj < 1 \times 10^{-3}$), CCA ($p\_adj < 1 \times 10^{-4}$), and DeepCorrSurv ($p\_adj < 1 \times 10^{-3}$). Comparisons with lmQCM showed consistent numerical improvements, but statistical significance was borderline in some cohorts ($p\_adj \approx 0.009$–$0.012$) and therefore did not uniformly meet the strict Bonferroni threshold.

OSCCA was the strongest baseline. MT-ADSCCA achieved consistent numerical improvements over OSCCA in BRCA, GBMLGG, and KIPAN, although the differences did not remain significant under the strict Bonferroni threshold (BRCA: $p\_adj = 0.018$; GBMLGG: $p\_adj = 0.036$; KIPAN: $p\_adj = 0.044$). The full paired test results for all feature-selection baselines are reported in Supplementary S4 Table. Table 2 summarizes the performance improvements over the best feature-selection baseline.

## Comparison with survival-model baselines (C-index)

To isolate the impact of MT-ADSCCA from downstream survival modeling choices, we compared survival prediction using the same MT-ADSCCA-selected feature set across four widely used survival frameworks: LASSO-Cox, Random Survival Forest (RSF), MTLSA, and DeepSurv. All models were evaluated using identical nested cross-validation folds.

As shown in Table 3, MT-ADSCCA achieved the highest C-index in all three cohorts. The largest improvement was observed in GBMLGG (+3.93% over DeepSurv), consistent with stronger nonlinear epigenetic interactions in glioma. Paired Wilcoxon signed-rank tests on per-fold C-index values showed that MT-ADSCCA significantly outperformed all survival baselines after Bonferroni correction across four comparisons within each cohort ($\alpha = 0.0125$): MT-ADSCCA vs. LASSO-Cox ($p\_adj < 1e-5$), RSF ($p\_adj < 1e-6$), MTLSA ($p\_adj < 1e-4$), and DeepSurv ($p\_adj < 0.002$).

**Table 2. Performance summary of MT-ADSCCA versus the best feature-selection baseline (C-index).**

| Dataset | MT-ADSCCA | Best Baseline | ΔC-index | % Improvement |
|---|---|---|---|---|
| BRCA | 0.7391 | OSCCA (0.7124) | +0.0267 | +3.75% |
| GBMLGG | 0.8449 | OSCCA (0.8289) | +0.0160 | +1.93% |
| KIPAN | 0.7812 | OSCCA (0.7692) | +0.0120 | +1.56% |

**Table 3. C-index comparison among survival-model baselines using the same MT-ADSCCA-selected feature set.**

| Dataset | LASSO-Cox | RSF | MTLSA | DeepSurv | MT-ADSCCA |
|---|---|---|---|---|---|
| BRCA | 0.6807 | 0.6523 | 0.6894 | 0.7087 | 0.7391 |
| GBMLGG | 0.7548 | 0.7426 | 0.7817 | 0.8056 | 0.8449 |
| KIPAN | 0.7342 | 0.7221 | 0.7413 | 0.7578 | 0.7812 |

## Risk stratification (Kaplan–Meier, log-rank, HR)

Using the predicted risk scores, patients were stratified into high- and low-risk groups using the median risk score within each test fold. Fig 1 shows the representative Kaplan–Meier curves for BRCA. MT-ADSCCA produced a clear separation with minimal overlap, and Table 4 summarizes the BRCA stratification performance across survival baselines. Across cohorts, MT-ADSCCA consistently yielded stronger survival separation and more favorable hazard ratio estimates than competing models, supporting its clinical relevance.

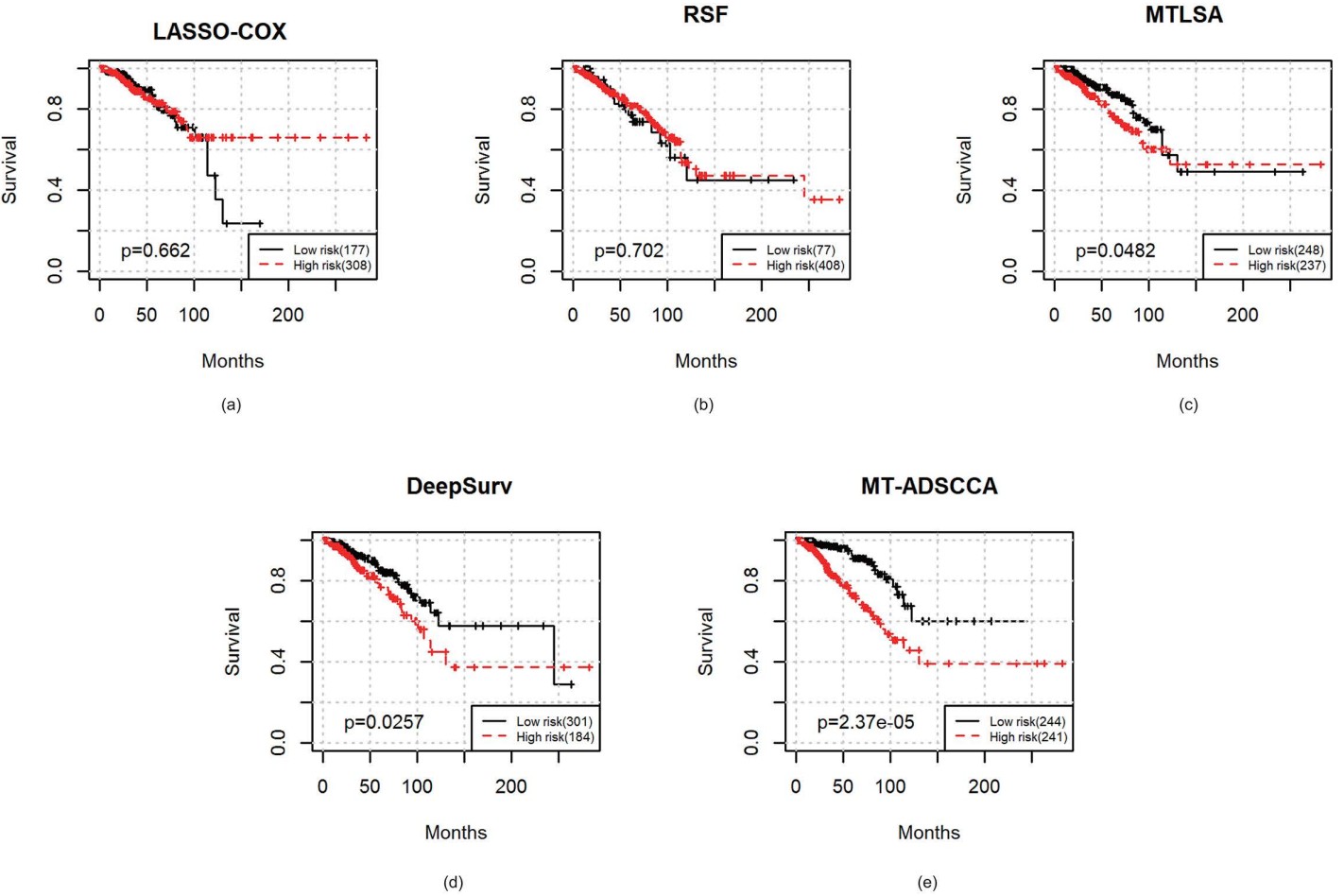

**Fig 1. The survival curves by applying different methods on BRCA datasets.**

**Table 4. BRCA example: Kaplan–Meier risk stratification performance across survival models.**

| Method | Log-rank p-value | HR (95% CI) |
|---|---|---|
| LASSO-Cox | 0.662 | 1.17 (0.88–1.56) |
| RSF | 0.702 | 1.20 (0.91–1.64) |
| MTLSA | 0.0482 | 1.62 (1.03–2.53) |
| DeepSurv | 0.00257 | 2.10 (1.43–3.29) |
| MT-ADSCCA | 2.37e−05 | 2.91 (1.92–4.17) |

## Shuffle test validating genomic ordering

To assess whether the BiLSTM–Cox survival head benefits from biologically meaningful feature ordering, we performed a shuffle test. Multi-omics features were ordered by chromosome and genomic coordinates before being passed into the BiLSTM–Cox head. We then randomly permuted the gene order 20 times per fold, retrained the survival head using identical hyperparameters, and evaluated the performance.

Shuffling consistently reduced the C-index and weakened KM separation across all cohorts (Table 5), demonstrating that genomic ordering contributes a meaningful structure rather than acting as an arbitrary sequence. The largest degradation occurred in GBMLGG, consistent with stronger spatial epigenetic dependencies in gliomas.

## Training dynamics: convergence and adaptive loss weights

Fig 2 summarizes the training dynamics of MT-ADSCCA on BRCA, GBMLGG, and KIPAN. Across cohorts, the total objective decreased sharply within the first 100–200 epochs and then stabilized, indicating reliable convergence of the multitask optimization. The correlation loss dropped most rapidly, suggesting that the deep encoders quickly captured nonlinear shared structure between methylation and expression. In contrast, the sparsity term converged more gradually and then plateaued, which is consistent with the formation of stable and compact biomarker signatures. No oscillations, divergences, or mode collapses were observed in any of the cohorts.

To verify the effectiveness of uncertainty-guided loss balancing, we tracked the learned task weights throughout training (Fig 3). The correlation-loss weight decreased moderately and stabilized around 0.82–0.90, whereas the sparsity-loss weight increased slightly, reflecting a progressive emphasis on interpretability after cross-omics alignment was established. Reconstruction and normalization weights remained stable or declined slightly, preventing auxiliary objectives from dominating the optimization. Importantly, none of the weights collapsed toward zero or diverged, supporting stable gradient magnitudes across tasks.

Finally, train-versus-validation canonical correlations remained closely aligned across folds (Supplementary S5 Table), indicating that MT-ADSCCA learned robust cross-omics representations without overfitting.

## Biological interpretation of selected canonical features

**Feature stability and selection reproducibility.** Specifically, the model selected 260 stable genes for BRCA, 1,003 for GBMLGG, and 576 for KIPAN. Selection frequency analysis demonstrated that more than 85% of selected BRCA

**Table 5. Shuffle test validating the genomic ordering used by the BiLSTM–Cox survival head.**

| Dataset | C-index drop after shuffling | Original log-rank p-value | Shuffled log-rank p-value (range) |
|---|---|---|---|
| BRCA | −2.8% | 2.37e−05 | 0.031–0.12 |
| GBMLGG | −5.4% | 6.40e−07 | 0.008–0.045 |
| KIPAN | −3.9% | 4.95e−04 | 0.017–0.083 |

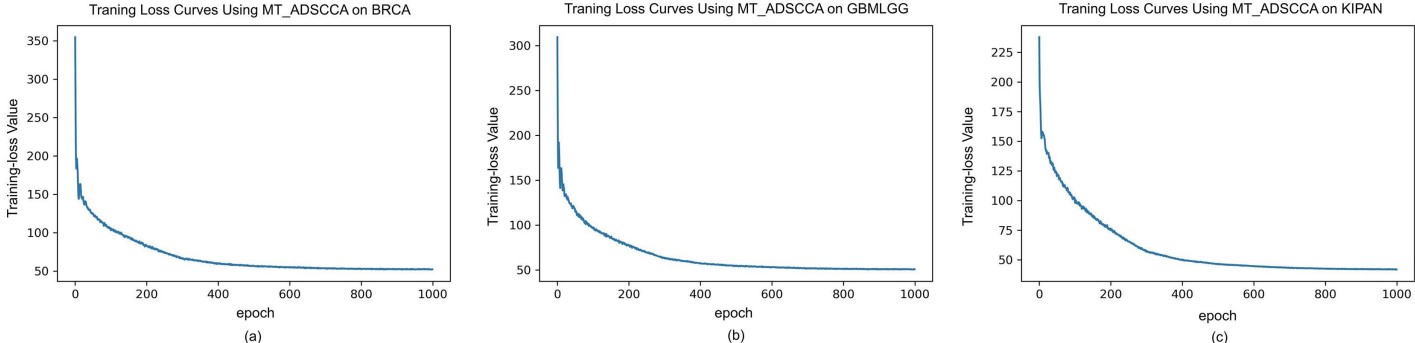

**Fig 2. The training loss curves using MT-ADSCCA methods on three datasets.** (a) the curves of training loss on BRCA dataset; (b) the curves of training loss on GBMLGG dataset; and (c) the curves of training loss on KIPAN dataset.

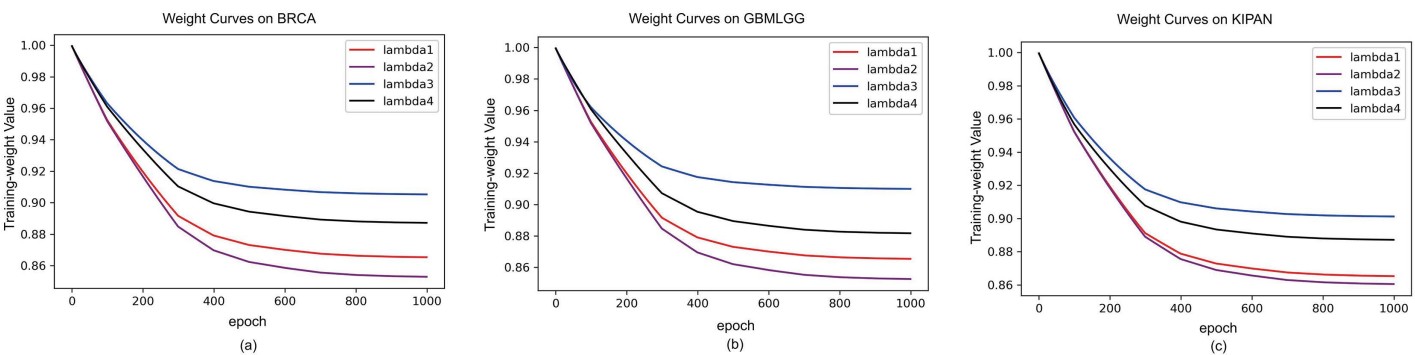

**Fig 3. The curves of weight variablesλ_i for the auxiliary task losses.** (a) the curves of weight variables on BRCA dataset; (b) the curve on GBMLGG dataset; and (c) the curve on KIPAN dataset.

genes, 78% of GBMLGG genes, and 82% of KIPAN genes appeared in ≥9 folds, indicating strong reproducibility despite high dimensionality and biological heterogeneity. This stability is a direct consequence of the model's multi-task sparsity enforcement combined with adaptive balancing of loss components.

**Prognostic relevance and top canonical genes.** To quantify survival relevance, we conducted univariate Cox regression for each selected gene. Fig 4 shows the 20 genes with the most significant Cox p-values in BRCA, GBMLGG, and KIPAN. The distribution of Cox p-values demonstrates that MT-ADSCCA systematically selects features with statistically meaningful associations with survival outcomes across all three cohorts.

**Functional-category enrichment and rationale for GO-based interpretation.** Conventional pathway-level enrichment (e.g., KEGG cell-cycle, DNA replication, or PI3K–Akt pathways) was not applied, as the MT-ADSCCA-selected features were not dominated by classical proliferation or DNA-repair hub genes such as CCNB1, MKI67, BRCA1/2, IDH1, or VHL. Under such conditions, KEGG enrichment can lead to unstable or misleading overinterpretation. Instead, functional-category enrichment based on GO Biological Process and protein family/domain categories was used, as it more accurately reflects the composition of the selected multi-omics features.

To interpret the canonical variates, we ranked genes by the absolute magnitude of their canonical coefficients ($|u|$ for mRNA expression and $|v|$ for DNA methylation) and performed univariate Cox regression on the MT-ADSCCA–selected features. Across all three cancer types (BRCA, GBMLGG, and KIPAN), the model consistently identified compact, biologically coherent, and tumor-type–specific multi-omics signatures rather than generic proliferation markers. To better

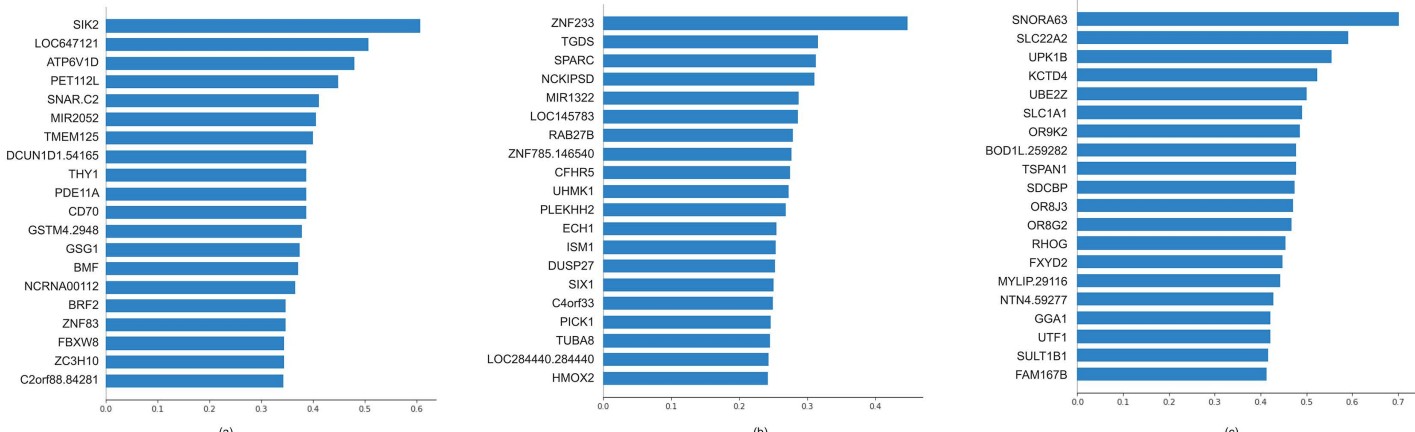

**Fig 4. Twenty most important genes for the survival models on three datasets.** (a) summary plots for importance on BRCA dataset; (b) summary plots for importance on GBMLGG dataset; and (c) summary plots for importance on KIPAN dataset.

understand the biological significance of the features selected by MT-ADSCCA, we conducted a detailed interpretation of canonical coefficients, examined univariate Cox associations, and analyzed functional categories enriched among the highest-weight genes.

## Discussion

In this study, we developed MT-ADSCCA, a multitask adaptive deep sparse canonical correlation framework designed to integrate DNA methylation and mRNA expression data for robust cancer survival prediction and interpretable biomarker discovery. Across three TCGA cohorts (BRCA, GBMLGG, and KIPAN), MT-ADSCCA consistently outperformed established feature selection algorithms and survival models. Beyond numerical improvements, MT-ADSCCA identified biologically coherent multi-omics signatures and produced stable risk stratification, underscoring its value for both computational methodology and translational biomarker discovery.

### Nonlinear and sparse canonical representations for multi-omics integration

Sparse CCA provides a principled approach for identifying correlated projections between paired omics layers; however, classical formulations are limited to linear relationships and typically require manual tuning of sparsity penalties. These constraints are particularly restrictive in cancer genomics, where methylation–expression relationships are often non-linear, context-dependent, and shaped by the tumor microenvironment and epigenetic remodeling. MT-ADSCCA addresses these limitations by combining nonlinear deep encoders with sparse canonical weights. The encoders learn flexible cross-omics transformations, whereas sparsity preserves interpretability by maintaining a direct mapping between canonical coefficients and individual genes.

The multitask loss design further stabilizes representation learning by jointly optimizing correlation maximization, sparsity enforcement, reconstruction consistency, and latent-space normalization. The adaptive uncertainty-based weighting mechanism reduces the sensitivity to manually selected loss coefficients and encourages balanced optimization across objectives. Together, these design choices support stable convergence and improved generalization in high-dimensional settings.

### Biological interpretation of selected canonical features

A central requirement for clinical multi-omics models is that predictive gains should correspond to biologically meaningful signals, rather than artifacts of high-dimensional optimization. In the Results section, we therefore emphasized

interpretation using functional categories derived directly from the observed top-ranked features, rather than imposing pre-defined hallmark gene sets. Across cohorts, MT-ADSCCA consistently recovered tumor-type-specific biological programs.

In BRCA, the most influential features were enriched for immune-related signaling, chromatin remodeling, and RNA helicase–associated transcriptional regulation, consistent with the immune-epigenetic interactions and regulatory plasticity that contribute to breast cancer heterogeneity. In GBMLGG, the dominant signals involved zinc-finger transcriptional regulation, extracellular matrix and adhesion processes, and metabolic remodeling, which align with known glioma microenvironmental dependencies and epigenetic reprogramming. In KIPAN, the top functional categories included solute and ion transport, ubiquitin-mediated proteostasis, and small GTPase signaling, reflecting metabolic transport and cytoskeletal regulation commonly implicated in renal cancer progression. These patterns support the nothion that MT-ADSCCA extracts coherent multi-omic signatures that are specific to tumor lineage, rather than converging on generic proliferation markers.

### Genomically ordered sequence modeling improves survival prediction

The validity of applying an LSTM-based survival head to gene-level features was assessed using a shuffle test (Supplementary S2 Table). By ordering multi-omics features according to chromosomal coordinates, BiLSTM can exploit spatially structured dependencies, such as local co-expression, co-methylation domains, and cis-regulatory neighborhoods. Random permutation of gene order led to a consistent decrease in the C-index and weaker Kaplan–Meier separation, indicating that genomic ordering provides a meaningful structure rather than an arbitrary sequence. This result supports the architectural choice and suggests that sequence modeling can capture biologically grounded neighborhood effects along the genome sequence.

### Comparison with existing feature-selection and survival baselines

MT-ADSCCA achieved consistently higher C-index values than six representative feature-selection baselines (DA, WGCNA, lmQCM, CCA, OSCCA, DeepCorrSurv) and four survival models (LASSO-Cox, RSF, MTLSA, DeepSurv). The strongest competitor among the feature-selection methods was OSCCA, which reflected the value of structured sparsity. However, MT-ADSCCA further improved performance through nonlinear correlation learning and adaptive multitask balancing. DeepSurv performed competitively among survival baselines, but it does not explicitly enforce sparse biomarker selection or cross-omics correlation structure, which limits interpretability. These trends were consistent with the paired statistical test results reported in Supplementary S4 Table.

### Clinical implications

MT-ADSCCA produces compact and interpretable multi-omics signatures and risk scores that stratify patients into clinically meaningful subgroups. The selected features highlighted immune–epigenetic interactions in BRCA, transcriptional regulation and extracellular matrix remodeling in glioma, and transporter- and signaling-driven programs in kidney cancers. These processes are directly relevant to therapeutic decision-making, including immunomodulatory strategies, epigenetic therapies, and treatments targeting tumor metabolism or microenvironmental dependencies. Because sparsity is explicitly enforced, MT-ADSCCA yields biomarker panels that are more amenable to downstream experimental validation and potential clinical assay development than dense deep learning representations.

### Limitations

This study had several limitations. First, the current implementation integrates two omics layers (mRNA expression and DNA methylation), reflecting modalities with the most complete overlap across TCGA cohorts. Although MT-ADSCCA can be generalized to more than two views, future studies should incorporate additional modalities, such as copy number variation, proteomics, and spatial transcriptomics. Second, despite multitask regularization and nested cross-validation,

high-dimensional multi-omics modeling remains vulnerable to cohort-specific effects, and external validation in independent datasets would further strengthen its clinical robustness. Third, although genomic ordering improves the BiLSTM head, alternative architectures, such as graph neural networks or attention-based models, may capture long-range regulatory dependencies more explicitly. Finally, the computational cost is higher than that of linear CCA-based methods, which may limit the scalability in ultra-large cohorts.

## Future directions

Future research will focus on extending MT-ADSCCA to a hierarchical multi-view framework capable of integrating more than two omics layers simultaneously, and on incorporating biological priors such as regulatory networks or chromatin interaction maps. Attention-based survival models may further improve the capture of long-range genomic dependency. Translationally, the identified biomarker panels should be validated experimentally and evaluated in prospective clinical settings to assess their utility in patient stratification and treatment selection.

## Conclusion

This study presents MT-ADSCCA, a multitask adaptive deep sparse canonical correlation framework for integrative cancer survival modeling using DNA methylation and mRNA expression data. By coupling nonlinear cross-omics representation learning with explicit sparsity, MT-ADSCCA bridges a key gap between deep multi-view learning and clinically interpretable biomarker discovery. Across three TCGA cohorts (BRCA, GBMLGG, and KIPAN), MT-ADSCCA consistently improved the survival prediction performance and generated compact multi-omics signatures that remained stable across cross-validation folds.

From a biomedical perspective, MT-ADSCCA highlights tumor-type-specific functional programs associated with survival heterogeneity. Rather than converging on generic proliferation markers, the selected features reflected coherent biological themes, including immune–epigenetic regulation in breast cancer, transcriptional control and extracellular matrix remodeling in glioma, and transporter-, ubiquitination-, and small GTPase–related mechanisms in kidney cancers. These signatures provide testable hypotheses for downstream mechanistic studies and support the potential development of reduced biomarker panels for clinical risk stratification.

From a machine learning perspective, MT-ADSCCA introduces an outcome-relevant multi-task formulation of sparse CCA with uncertainty-guided adaptive loss weighting, improving training stability and reducing dependence on manual loss tuning. The framework also integrates a genomically ordered BiLSTM–Cox survival head, and shuffle testing demonstrated that genomic ordering contributes a meaningful structure for risk prediction. Together, these components provide a practical strategy for extracting interpretable nonlinear cross-omics dependencies while maintaining a strong predictive performance.

In summary, MT-ADSCCA offers a unified and reproducible approach for multi-omics integration, sparse biomarker identification, and survival prediction. Future work will extend the framework to additional omics layers and external validation cohorts and explore alternative biologically informed architectures to capture long-range regulatory dependencies. Ultimately, MT-ADSCCA provides a step toward clinically actionable precision oncology models that are both accurate and interpretable.

## Supporting information

**S1 Table. Nested cross-validation performance (outer test folds).** Metric: concordance index (C-index). Outer CV = 10 folds; inner CV = 10 folds for tuning. Values are mean ± SD across outer folds.
(DOCX)

**S2 Table. Shuffle test for the genomically ordered BiLSTM–Cox survival head.** Genes were ordered by chromosome and genomic coordinate in the original model. In the shuffled setting, gene order was randomly permuted 20 times per fold; the table reports mean ± SD across all permutations and folds.
(DOCX)

**S3 Table. DeepSurv baseline configuration used in this study.** This configuration was fixed across cohorts and tuned within the inner cross-validation loop where applicable.
(DOCX)

**S4 Table. Additional multi-omics integration baselines under identical folds.** Latent representations from each method were evaluated using the same survival head for fairness. Statistical tests were performed on per-fold C-index values (paired Wilcoxon). Bonferroni correction was applied within each cohort.
(DOCX)

**S5 Table. Canonical correlation of the first canonical pair (train vs validation).** Values are mean±SD across outer folds. Corr (zx1, zy1) denotes the canonical correlation between the first latent components learned from expression and methylation encoders.
(DOCX)

## Author contributions

**Conceptualization:** Yan Wang, Zimo Zou.

**Data curation:** Yan Wang, Zimo Zou, Yuanyuan Wu.

**Formal analysis:** Zimo Zou, Yuanyuan Wu.

**Funding acquisition:** Yan Wang.

**Investigation:** Yan Wang, Zimo Zou, Yuanyuan Wu.

**Methodology:** Yan Wang, Zimo Zou, Yuanyuan Wu.

**Project administration:** Yan Wang.

**Software:** Zimo Zou.

**Supervision:** Yan Wang.

**Validation:** Yan Wang, Zimo Zou, Yuanyuan Wu.

**Visualization:** Zimo Zou, Yuanyuan Wu.

**Writing – original draft:** Yan Wang, Zimo Zou.

**Writing – review & editing:** Yuanyuan Wu, Jing Chen.

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
