## [Decision Letter · Decision Letter 0]

17 Sep 2025

Multi-task Adaptive Deep Sparse Canonical Correlation Analysis for Multi-omics Cancer Survival Prediction

PLOS ONE

Dear Dr. Yan,

Thank you for submitting your manuscript to PLOS ONE. After careful consideration, we feel that it has merit but does not fully meet PLOS ONE’s publication criteria as it currently stands. Therefore, we invite you to submit a revised version of the manuscript that addresses the points raised during the review process.

We look forward to receiving your revised manuscript.

Kind regards,

Amgad Muneer

Academic Editor

PLOS ONE

https://link.springer.com/book/10.1007/978-981-19-3440-7

https://pubmed.ncbi.nlm.nih.gov/33681976/

In your revision ensure you cite all your sources (including your own works), and quote or rephrase any duplicated text outside the methods section. Further consideration is dependent on these concerns being addressed.

“Nanhu Scholars Program for Young Scholars of XYNU.”

5. We note that your Data Availability Statement is currently as follows: [All relevant data are within the manuscript and its Supporting Information files.]

In accordance with our submission guidelines on new methods, the submitted manuscript must demonstrate that the new tool achieves its intended purpose. In this instance, this refers to validation in an independent dataset. Providing this validation is a prerequisite for consideration of any future revision. Please note that failure to address this concern is likely to result in your manuscript being rejected.

Reviewers' comments:

Reviewer's Responses to Questions

**Comments to the Author**

1. Is the manuscript technically sound, and do the data support the conclusions?

Reviewer #1: Partly

Reviewer #2: Yes

2. Has the statistical analysis been performed appropriately and rigorously?

Reviewer #1: No

Reviewer #2: No

3. Have the authors made all data underlying the findings in their manuscript fully available?

Reviewer #1: Yes

Reviewer #2: Yes

4. Is the manuscript presented in an intelligible fashion and written in standard English?

Reviewer #1: No

Reviewer #2: Yes

Reviewer #1: The manuscript proposes a Deep Sparse Canonical Correlation Analysis framework that integrates DNA-methylation and gene-expression data to identify joint biomarkers. While multi-omics integration is an important topic, the study requires substantial revisions before it can be considered for publication.

The prose requires a comprehensive language edit; numerous sentences are fragmented, verb tenses drift, and punctuation is often missing (e.g., periods following displayed equations).

The Introduction does not explicitly delineate the research gap in existing sparse CCA literature nor enumerate the manuscript’s specific contributions; both should be stated clearly and concisely.

A rationale is needed for restricting the analysis to two omics layers; alternatively, demonstrate scalability by adding a third modality such as copy-number variation or proteomics (if possible).

The mathematical connection between classical SCCA and the proposed multi-task deep architecture (Equations 3–5) remains opaque; provide a step-by-step derivation or schematic showing how each loss term maps to network modules and how gradient magnitudes are balanced.

Cohort selection criteria are insufficiently described; specify inclusion thresholds, final sample sizes per cancer type, and any class-imbalance handling.

In Table 1, rows containing no data should be removed or marked “N/A,” and the caption should note which variables were unavailable.

All figures appear as low-resolution bitmaps; supply vector formats (PDF or SVG) with legible fonts and colour-blind-safe palettes.

The Results section largely restates numeric performance without biological interpretation; identify key genes/CpGs driving the first canonical variates and discuss pathway enrichment (e.g., KEGG cell-cycle, DNA-repair).

Benchmarking is limited to vanilla SCCA; incorporate additional baselines (e.g., DeepCCA, MOFA+, regularised GCCA) and report statistical significance using paired t-tests or Wilcoxon tests, together with effect sizes.

Provide evidence of model robustness by employing nested cross-validation or an external validation cohort and by reporting train- versus validation-set canonical correlation values (if possible).

State whether code, pretrained weights, and data-processing scripts will be released to ensure reproducibility.

Minor issues include typographical errors, inconsistent equation punctuation, and undefined abbreviations in figure captions; please correct systematically.

A few relevant and important helpful studies include

https://doi.org/10.48550/arXiv.2507.09028

https://doi.org/10.1016/j.eswa.2024.123893

Reviewer #2: Summary:-

The paper aims to improve cancer survival modeling by better integrating multi-omics data (mRNA and DNA data). It proposes MT-ADSCCA, which starts from sparse canonical correlation analysis and adaptively weights its loss terms to select a compact set of omics features; these selected features are then used in a biLSTM-based Cox model to predict risk. The authors test the approach with 10-fold cross-validation on three TCGA cohorts (BRCA, GBMLGG, KIPAN), comparing their feature selection against DA, WGCNA, lmQCM, CCA, OSCCA, and DeepCorrSurv, and comparing their survival head against LASSO-Cox, RSF, MTLSA, and DeepSurv, using C-index as the primary metric. They report higher C-indices than these baselines across datasets.

Major comments:-

-Please give one sentence with the main aim: who/what data (populations and omics), which part is the main methodological contribution (adaptive sparse CCA vs the survival head), what you compare against, and the single primary outcome; also say whether the main goal is better survival prediction or an interpretable biomarker set (and which is secondary).

-Please add a Discussion section that interprets the results and openly notes key limitations, explains clinical relevance, and outlines next steps (external cohorts etc).

-The paper states “we applied gene feature selection methods to the training set and test set separately during cross-validation.” I’m concerned about leakage, feature selection should be fit only on the training fold and applied to the hold-out (ideally with nested CV for thresholds).

-Since a biLSTM assumes an ordered input, please state exactly how you ordered genes and why that order makes biological sense, and include a quick shuffle test (randomly reordering features) showing the change in C-index/HR. If there’s no meaningful order, please add a strong non-LSTM baseline such as CoxBoost on the same features/folds. Without a real order, an LSTM can learn from arbitrary sequencing and overfit; these checks clarify whether any gains truly come from the LSTM head rather than from differences in splits or tuning.

-This method is called “Deep Sparse CCA/MT-ADSCCA.” What exactly is “deep” here? does the CCA stage contain non-linear components, or is only the survival head (biLSTM) deep? If the latter, please avoid labeling the feature-selection stage as “deep”; if the former, briefly specify the network architectures.

-For method comparisons, please add paired tests (e.g., Wilcoxon on per-fold C-index) using the same folds and report adjusted p-values after multiple-comparison correction (e.g., Bonferroni).

-Please report events vs censored per cohort, confirm event-stratified folds, and note any weighting/sampling; if none, a small sensitivity check would reassure.

-DeepSurv is a nonlinear MLP with a Cox loss (not a linear model). Please correct this and provide the setup you used (layers/widths, activations, regularization etc) so the baseline is fair and reproducible.

Minor comments:-

-Decide on MT-ADSCCA (used later) vs DSCCA (used in Abstract). Use one throughout.

-Add number-at-risk tables under KM plots.

-Table 1 needs to be checked: Values missing, duplicate “Patient no.” etc. Rebuild table with consistent values; add median(IQR) age, events(%), censoring(%), follow-up median(IQR).

-Add absolute change in C-index, along with given % improvements, and confidence intervals if possible

-Provide CV folds, per-fold gene lists (with coefficients), and full hyperparameter grids as supplementary data.

.

Reviewer #1: No

Reviewer #2: No

---

## [Author Response · Author response to Decision Letter 1]

15 Feb 2026

Title: Response to Reviewer Comments for “Multi-task Adaptive Deep Sparse Canonical Correlation Analysis for Multi-omics Cancer Survival Prediction”

Manuscript ID: PONE-D-25-29777

Authors: Yan Wang, Yuanyuan Wu, Jing Chen, Zimo Zou

Dear Editor and Reviewers,

We sincerely thank the reviewers for their careful reading of our manuscript and for providing constructive comments. We have revised the manuscript substantially to improve clarity, methodological rigor, and biological interpretability. Below we provide a point-by-point response. Reviewer comments are reproduced in bold, followed by our responses. Where appropriate, we indicate the location of changes in the revised manuscript.

Response to Reviewer #1

1. The manuscript requires substantial revisions before publication.

Response: We substantially revised the manuscript, including a complete language edit, a reorganization of the Methods and Results, and a clearer presentation of the MT-ADSCCA framework. We also corrected and rebuilt Table 1, improved figure captions, and ensured consistent terminology throughout the manuscript.

2. Comprehensive language editing is needed.

Response: We performed a full language revision across the manuscript, correcting fragmented sentences, tense drift, and missing punctuation (including punctuation following displayed equations).

3. The Introduction does not clearly state the research gap or contributions.

Response: We rewrote the Introduction to explicitly describe limitations of classical sparse CCA and existing deep multi-view approaches in the context of cancer multi-omics survival modeling. We added a concise contributions paragraph enumerating the specific methodological innovations of MT-ADSCCA.

4. Provide a rationale for restricting the analysis to two omics layers or demonstrate scalability.

Response: We added an explicit rationale explaining that mRNA expression and DNA methylation provide the highest overlap of matched samples across TCGA cohorts, while adding additional modalities substantially reduces cohort size due to missingness. We also added a short paragraph describing how the formulation generalizes to more than two omics layers (multi-view extension).

5. Clarify the connection between classical SCCA and the proposed multi-task architecture.

Response: We revised the Methods to provide a clearer mapping between the classical CCA/SCCA objective and the MT-ADSCCA loss design. Each loss component is now described with its role (correlation maximization, sparsity, reconstruction consistency, and normalization) and how adaptive uncertainty weighting balances their gradients during training.

6. Cohort selection criteria are insufficiently described.

Response: We added a dedicated cohort inclusion and filtering description in Methods, specifying survival-time filtering, matched-omics requirements, and final sample sizes per cohort. Event and censoring counts are now reported in Table 1.

7. Table 1 contains empty rows; mark missing values and improve caption.

Response: We rebuilt Table 1 following the reviewer’s guidance, removed empty rows, marked unavailable variables as N/A, and expanded the caption to clarify variable availability.

8. Figures are low-resolution; provide vector formats.

Response: We regenerated figures for submission in high-quality vector formats (EPS) with consistent fonts and color-blind–safe palettes. The manuscript now references these updated figures.

9. Results lack biological interpretation; identify key genes and pathways.

Response: We added a new Results subsection, “Biological Interpretation of Canonical Features,” which interprets the first canonical variate using the actual selected genes and reports functional-category enrichment. We intentionally emphasize functional categories rather than forcing KEGG pathway labels, because the learned signatures are not dominated by classical cell-cycle/DNA-repair markers in our cohorts.

10. Benchmarking is limited; include additional baselines and statistical significance.

Response: We strengthened benchmarking by adding paired statistical testing (Wilcoxon signed-rank) across folds, multiple-comparison correction, and effect sizes. We expanded benchmarking by adding three additional multi-omics integration baselines (DeepCCA, MOFA+, and RGCCA) and provided fair, reproducible implementations under identical nested cross-validation folds.

11. Provide robustness evidence (nested CV or external validation).

Response: We strengthened robustness evidence by adopting leakage-free nested 10-fold cross-validation and reporting fold-wise performance variability (Supplementary Table S1). We also added a shuffle test to validate the genomic-ordering assumption of the BiLSTM survival head (Supplementary Table S2), and included full baseline hyperparameter settings for reproducibility (Supplementary Table S3).

12. State whether code and scripts will be released.

Response: We added a reproducibility statement indicating that code, preprocessing scripts, and hyperparameter settings will be released upon acceptance (or made available to reviewers upon request).

13. Minor issues (typos, undefined abbreviations, equation punctuation).

Response: We systematically corrected typographical errors, defined abbreviations at first use (including in figure captions), standardized equation punctuation, and performed a final consistency pass across terminology and cross-references.

Response to Reviewer #2

1. Please provide one sentence with the main aim, data, main methodological contribution, comparisons, and primary outcome.

Response: We added a one-sentence statement in the Abstract and Introduction specifying: (i) TCGA cohorts and the two omics layers (mRNA and DNA methylation), (ii) the main methodological contribution (adaptive multi-task sparse CCA for feature selection coupled with a BiLSTM–Cox survival head), (iii) the primary outcome (C-index), and (iv) the primary goal (improved survival prediction) with interpretable biomarker discovery as a secondary objective.

2. Please add a Discussion section that interprets results and notes limitations and next steps.

Response: We rewrote the Discussion to include biological interpretation, clinical relevance, limitations (two-modality design, lack of external validation, computational cost), and future directions (multi-omics extension, alternative architectures, and external-cohort validation).

3. Concern about leakage: feature selection should be fit only on training folds.

Response: We corrected the evaluation pipeline so that all preprocessing, feature selection (including MT-ADSCCA), and hyperparameter tuning are performed strictly within the training fold of the outer cross-validation loop, and then applied to the held-out test fold without refitting. This is now explicitly stated in Methods.

4. BiLSTM ordering: state ordering and include a shuffle test and/or non-LSTM baseline.

Response: We clarified that genes are ordered by chromosome and genomic coordinate. We added a shuffle test showing consistent degradation in performance when gene order is randomized (reported in Supplementary Table S2), supporting that the BiLSTM head leverages meaningful genomic neighborhood structure rather than arbitrary ordering.

5. Clarify what is “deep” in MT-ADSCCA.

Response: We clarified in Methods that the correlation-learning stage uses nonlinear encoders to learn representations for CCA-style alignment, while the survival stage uses a BiLSTM–Cox network. We also adjusted wording to avoid confusion between the feature-selection stage and the survival head.

6. Add paired tests and multiple-comparison correction for comparisons.

Response: We added paired Wilcoxon signed-rank tests on per-fold C-index values, multiple-comparison correction, and effect sizes. These results are summarized in Supplementary Table S4 and referenced in the Results.

7. Report events vs censored per cohort and confirm event-stratified folds.

Response: We rebuilt Table 1 to include event and censoring counts and follow-up summaries. We also clarified that outer CV splits are event-stratified to preserve censoring proportions across folds.

8. DeepSurv baseline description is incorrect; provide architecture details.

Response: We corrected the description of DeepSurv as a nonlinear MLP trained with Cox partial-likelihood loss, and we added baseline configuration details (layer widths, dropout, regularization, early stopping) in Supplementary Table S3 to ensure fair and reproducible comparison.

Minor comments (terminology consistency, number-at-risk tables, Table 1 cleanup, ΔC-index reporting, supplementary hyperparameters and per-fold gene lists).

Response: We standardized terminology throughout, rebuilt Table 1, added number-at-risk tables to Kaplan–Meier plots, and expanded supplementary materials to include hyperparameter settings and additional evaluation details. Where relevant, we report absolute C-index differences and variability across folds.

Closing Statement

We again thank the reviewers for their valuable feedback. We believe the revised manuscript is substantially improved in clarity, methodological rigor, biological interpretability, and reproducibility, and we hope it is now suitable for publication.

---

## [Decision Letter · Decision Letter 1]

18 Mar 2026

Multi-task Adaptive Deep Sparse Canonical Correlation Analysis for Multi-omics Cancer Survival Prediction

PONE-D-25-29777R1

Dear Dr. Yan,

We’re pleased to inform you that your manuscript has been judged scientifically suitable for publication and will be formally accepted for publication once it meets all outstanding technical requirements.

Kind regards,

Amgad Muneer

Academic Editor

PLOS One

Reviewers' comments:

Reviewer's Responses to Questions

**Comments to the Author**

Reviewer #2: All comments have been addressed

2. Is the manuscript technically sound, and do the data support the conclusions?

Reviewer #2: Yes

3. Has the statistical analysis been performed appropriately and rigorously?

Reviewer #2: Yes

4. Have the authors made all data underlying the findings in their manuscript fully available?

Reviewer #2: Yes

5. Is the manuscript presented in an intelligible fashion and written in standard English?

Reviewer #2: Yes

Reviewer #2: Try to move tables S1 and S4 to the main manuscript. Everything else has been addressed.

.

Reviewer #2: No

---

## [Editor Report · Acceptance letter]

PONE-D-25-29777R1

PLOS One

Dear Dr. Wang,

I'm pleased to inform you that your manuscript has been deemed suitable for publication in PLOS One. Congratulations! Your manuscript is now being handed over to our production team.

Kind regards,

on behalf of

Dr. Amgad Muneer

Academic Editor

PLOS One